# A Deep Learning Approach to Classify Surgical Skill in Microsurgery Using Force Data from a Novel Sensorised Surgical Glove

**DOI:** 10.3390/s23218947

**Published:** 2023-11-03

**Authors:** Jialang Xu, Dimitrios Anastasiou, James Booker, Oliver E. Burton, Hugo Layard Horsfall, Carmen Salvadores Fernandez, Yang Xue, Danail Stoyanov, Manish K. Tiwari, Hani J. Marcus, Evangelos B. Mazomenos

**Affiliations:** 1Wellcome/EPSRC Centre for Interventional and Surgical Sciences, University College London, London W1W 7TY, UK; jialang.xu.22@ucl.ac.uk (J.X.); dimitrios.anastasiou.21@ucl.ac.uk (D.A.); james.booker.19@ucl.ac.uk (J.B.); ollieburton@doctors.org.uk (O.E.B.); hugo.layardhorsfall@ucl.ac.uk (H.L.H.); carmen.fernandez.15@ucl.ac.uk (C.S.F.); yang.xue.22@ucl.ac.uk (Y.X.); danail.stoyanov@ucl.ac.uk (D.S.); m.tiwari@ucl.ac.uk (M.K.T.); h.marcus@ucl.ac.uk (H.J.M.); 2Department of Medical Physics and Biomedical Engineering, University College London, London WC1E 6BT, UK; 3Victor Horsley Department of Neurosurgery, National Hospital for Neurology and Neurosurgery, London WC1N 3BG, UK; 4Nanoengineered Systems Laboratory, UCL Mechanical Engineering, University College London, London WC1E 7JE, UK; 5Department of Computer Science, University College London, London WC1E 6BT, UK

**Keywords:** surgical skill classification, microsurgery, neurosurgery, force data, deep learning

## Abstract

Microsurgery serves as the foundation for numerous operative procedures. Given its highly technical nature, the assessment of surgical skill becomes an essential component of clinical practice and microsurgery education. The interaction forces between surgical tools and tissues play a pivotal role in surgical success, making them a valuable indicator of surgical skill. In this study, we employ six distinct deep learning architectures (LSTM, GRU, Bi-LSTM, CLDNN, TCN, Transformer) specifically designed for the classification of surgical skill levels. We use force data obtained from a novel sensorized surgical glove utilized during a microsurgical task. To enhance the performance of our models, we propose six data augmentation techniques. The proposed frameworks are accompanied by a comprehensive analysis, both quantitative and qualitative, including experiments conducted with two cross-validation schemes and interpretable visualizations of the network’s decision-making process. Our experimental results show that CLDNN and TCN are the top-performing models, achieving impressive accuracy rates of 96.16% and 97.45%, respectively. This not only underscores the effectiveness of our proposed architectures, but also serves as compelling evidence that the force data obtained through the sensorized surgical glove contains valuable information regarding surgical skill.

## 1. Introduction

### 1.1. Background

Microsurgery uses miniaturized instruments and an operating microscope to achieve fine surgical tasks such as anastomosis of blood vessels or nerves. Microsurgery originated in the late 19th century in the field of vascular surgery, with the first end-to-end anastomosis of a blood vessel performed using a fine silk suture [1]. However, these initial surgeries had many complications due to the formation of thrombi during the procedure and poor visualization of tissues using crude magnification systems. The discovery of the anticoagulant heparin in 1916 and the development of the operating microscope in the 1920s revolutionized the success of microsurgery and expanded its application to other surgical fields including plastic surgery, ophthalmology, and neurosurgery [1]. Microsurgery now forms a key component of many operative procedures, but it has an abrupt learning curve for novice surgeons to achieve clinical competency [2].

Surgical skill level is an important determinant in reducing post-operative complications [3]. Surgical skills are traditionally developed through years of surgical training in an apprenticeship model. Trainee surgeons gradually advance their psychomotor surgical skills while moving towards independent practice. However, the intraoperative feedback provided by this model is highly variable, unstructured, and prone to bias [4]. The apprenticeship model also requires reflection on the part of the surgeon, and the learning from this can be limited by inaccurate self-assessment. Indeed, there has been a move in recent years away from the traditional apprenticeship model and towards competency-based surgical training [5]. It focuses on the development of a surgeon’s technical competencies and capabilities through regular summative assessments and not time in training [5]. Through this, surgical skills developed through simulated training have become a major component of surgical training and have been shown to transfer to performance intraoperatively [6].

To provide objective feedback to surgeons in training, many surgical skill assessment tools have been developed in microsurgery [7]. These assessment tools are designed to provide objective feedback to surgeons through assessments using surgical simulator models prior to exposure to real surgery. Anonymized assessments are then reviewed retrospectively by a senior surgeon to provide objective feedback. However, these assessment tools do not provide real-time feedback to surgeons and therefore cannot be used intraoperatively. Furthermore, they require an experienced microsurgeon to rate the participant which is time-consuming and costly, as microsurgery is a skill developed over many years of practice.

A potential solution to standardizing the development of surgical skills is to provide automated and objective feedback using machine learning (ML). At its core, surgery involves the manipulation of tissue to treat disease, primarily guided by force feedback. Novice surgeons are known to struggle with choosing the appropriate force to exert during surgical procedures and this is linked to over half of surgical errors [8].

Errors associated with inappropriate application of force during surgery include the unintended dissection of tissues or damage of blood vessels leading to bleeding [9]. Therefore, tasks that serve to assess competence (and distinguish expert surgeons from novices) should concern the consistent application of force throughout the task with little variability, such as the dissection of symmetrical structures, and where misapplied forces produce a distinguishable, displeasing or asymmetric result.

Tool–tissue interaction forces are known to be a key component of surgical success [10]. Indeed, novice surgeons are known to exert significantly higher forces and with greater force variability than expert surgeons [9]. Additionally, in previous studies, the force data were derived indirectly from robots or surgical instruments and not from the surgeons’ hands [11,12,13,14,15,16]. A novel approach developed by our group was the use of a sensorized surgical glove able to measure a range of intraoperative forces (0–10 N) and differentiate between novice and expert surgeons [17]. This sensorized glove is low in cost and all components can be heated to >200 °C, allowing for routine medical sterilization between surgical cases [18]. However, the potential of employing ML to analyze data from the sensorized glove for deeper insights into surgical skills remains unexplored. This study aims to leverage ML methods for the first time for classifying surgical skill levels using intraoperative force data from a microsurgical task. Such a system can facilitate surgical training and education at a low cost.

#### Problem Definition

This study centers on the application of deep learning techniques for the automated assessment of surgical skills based on force data in the context of microsurgery. Our primary objective is to treat skill assessment as a classification task, where the neural network classifies a surgical execution as either ‘Expert’ or ‘Novice’. To accomplish this, we conduct an extensive and in-depth comparison of various deep learning models, encompassing Long Short-Term Memory (LSTM), Gate Recurrent Unit (GRU), Bidirectional LSTMs, Temporal Convolutional Networks (TCN), Transformer and custom Convolutional Long Short-Term Memory Deep Neural Network (CLDNN) architectures.

With the aim of achieving robust performance, we undertake an exploration of data augmentation techniques, with a specific focus on our top-performing models. This investigation aims to determine whether data augmentation can mitigate the challenges posed by the relatively small dataset and enhance the models’ classification performance.

### 1.2. Related Work

Force exhibits a strong correlation with experience levels in minimally invasive surgery [19,20], making it a highly efficient and effective data source for automated surgical skill assessment [11,21]. Rosen et al. develop Markov models of forces and torques for skill assessment in laparoscopy [11]. Horeman et al. use principal components derived from force parameters like maximal absolute force, mean force, and force variability to distinguish between experts and novices, confirming the significant impact of force on suture tasks [21]. Brown et al. utilize contact forces and robot arm accelerations as inputs to a Random Forest classifier to predict skill levels for a peg transfer task [22]. Rafii-Tari et al. propose hidden Markov models based on operator motions and catheter–tissue interaction forces for skill assessment in robotic endovascular catheterization [23]. However, these approaches typically demand substantial time and computational resources for tasks such as feature engineering, parameter optimization, and model development [24]. Furthermore, they necessitate the collection of complete observations for each trial, which hinders the feasibility of conducting real-time skill assessments [25].

Recently, deep learning-based models have shown promise in surgical skill assessment. Fawaz et al. design a one-dimensional Convolutional Neural Network (CNN) based on kinematic data to assess skill levels on the suturing and needle-passing tasks of the JIGSAWS dataset [26]. Nguyen et al. employ accelerometer measurements and a neural network with parallel branches, combining a 1D CNN and an LSTM, to distinguish between three levels of surgical expertise [27]. Similarly, Wang and Majewicz Fey present an online skill assessment framework by utilizing 1D CNN to map multivariate motion kinematics data to individual skill levels [25]. Furthermore, Anh et al. provide a comprehensive summary and investigation of kinematics-based frameworks, including CNNs, Recurrent Neural Networks (RNNs), and autoencoders, all designed for real-time surgical skill assessment, with remarkable performance observed in suturing, knot-tying, and needle-passing tasks [28]. Other studies use image and video data to perform automated skill assessments. For example, Lajkó et al. introduce a 2D endoscopic image-based deep learning benchmark to enable the creation and application of surgical skill assessment in the minimally invasive surgery training environment [29]. Funke et al. propose a video-based surgical skill assessment framework by using an inflated 3D CNN to assign video segments to different skill levels [30]. Soleymani et al. propose a method using ResNet-50 for feature extraction from video data, followed by TCN for temporal modeling, achieving notable performance in surgical skill classification on the JIGSAWS dataset [31]. Furthermore, Kiyasseh et al. introduce a video-based framework for simultaneous recognition of skill levels, as well as surgical phases and gestures in Robot-Assisted Radical Prostatectomies, leveraging a pre-trained Vision Transformer in a self-supervised manner achieving robust performance across various experimental setups [32]. However, to the best of our knowledge, there is currently no established deep learning benchmark study for surgical skill assessment based on force data in the field of microsurgery.

### 1.3. Contributions

The main contributions of this paper include the following aspects:Our work represents the original effort to address Force-based automated Skill Classification (FSC) in microsurgery.We establish a benchmark for FSC, including six prominent deep learning architectures and six data augmentation techniques. This benchmark is accompanied by a comprehensive analysis, both in quantitative and qualitative aspects, to provide valuable insights for future research in this domain.We build a novel FSC dataset, comprising 236 trials conducted by 13 surgeons, which includes time-series force data with corresponding skill level annotations. Two dataset split settings, namely random split and Leave-One-User-Out (LOUO), are applied to train the proposed benchmark, presenting the first demonstration of how to use and evaluate this new dataset.Experimental results on the proposed dataset show that the temporal convolutional network with time warp data augmentation achieves the highest 97.46% accuracy for the random split setting, and the convolutional long short-term deep neural network with time warp data augmentation reaches the best 96.48% accuracy for the LOUO setting.

## 2. Materials and Methods

### 2.1. Dataset

#### 2.1.1. Surgical Task and Participants

The study recruited 13 surgeons from a single tertiary university hospital during surgical skill courses. Surgeons were categorised as ’Novice’ if they completed 5 surgical cases or ’Expert’ if they achieved the Certificate of Completion of Training (CCT) [33,34]. All surgeons in this study were right-handed. No power calculation was used, but the sample size was based on previous similar studies in the field [9,34,35,36,37]. The Strengthening the Reporting of Observational Studies in Epidemiology (STROBE) reporting guidelines were followed [38].

Participants were asked to wear the sensorized surgical glove with the piezoresistive foam sensor mounted on the thumb (Figure 1a) and to perform a validated preclinical microsurgical task ’Star’s the limit’ [17,39]. A standardized star was drawn on a grape using a stencil with a 5 mm edge length. Participants were required to incise within the black line of the drawn star and peel the star-shaped skin off the grape whilst minimizing damage to the grape flesh (Figure 2). Microscissors and forceps were provided to the participants. Each participant repeated the task 20 times. The OPMI PENTERO or KINEVO 900 (Carl Zeiss Co., Ltd., Jena, Germany) operating microscopes were used with focal lengths of 200–500 mm and 200–625 mm, respectively. Surgeons were blinded to the real-time forces because this was a proof-of-concept study examining the feasibility of sensorized surgical gloves. No feedback or teaching was provided to the participants during the task. If participants were not able to finish within 5 min, they were told to stop, and the next repetition would begin.

#### 2.1.2. Hardware Description

#### Materials

Carbon nanofiber (CNF) (D × L—100 nm × 20–200 µm, Sigma Aldrich, St. Louis, MO, USA), Copper tape (10 mm × 33 m, RS Components Ltd., Northamptonshire, UK), Ethanol (absolute, Sigma Aldrich), Instant adhesive (Loctite 401, Henkel Adhesives, Düsseldorf, Germany), Melamine foam (Doktor Power Ltd., JML, London, UK), Polyimide (kapton) tape (2.54 cm width, Cole-Parmer, Vernon Hills, IL, USA), and Tannic acid (Sigma Aldrich) were all used as received.

#### Sensor Fabrication

The design of these sensors was aimed towards achieving cost efficiency, ease of manufacturing, customisability, softness and flexibility. To meet these requirements, a conductive porous foam (melamine) was used as the sensing element. The conductive foam consists of conducting nanofibres embedded in a porous framework. When pressure is applied to the foam, the conducting nanofibres within the foam move closer together, thus increasing the material’s electrical conductivity [35].

Sensor fabrication consisted in first dipping the coating in a suspension of electrically conducting CNF into the interconnected microporous framework of melamine foam. CNF (0.6 wt.%) was dispersed in a tannic acid (0.5 wt.%) solution in distilled water by bath sonicating (FB15053, 200 W, Fisher, Hampton, NH, USA) for 20 min. Next, melamine foam (precleaned by washing with ethanol) was cut into any desired dimensions, for instance, those that fit a fingertip of the glove (L × W × H: 10 mm × 8 mm × 3 mm) and soaked into the carbon nanofiber suspension for 2 min before drying in an oven at 80 °C. The dried foams were then sandwiched between two copper foils attached to wires serving as the electrodes. A complete piezoresistive sensor was obtained after packaging with polyimide (kapton) tapes [35].

#### Sensorized Glove Fabrication

The piezoresistive sensors were directly mounted onto the fingertips of surgical gloves (Figure 1) using an instant adhesive (Loctite 401) along with the thin silicone wires (TUOFENG 30 AWG, 0.05 mm${}^{2}$) connected to the electrodes.

#### Sensor Calibration

A force calibration setup was developed to convert the resistance measurements from the piezoresistive sensors into force readings. The test setup and protocol used a motorised translation stage (PT1-Z8, Thorlabs, Newton, NJ, USA), a force gauge (M5-5 Mark10) and a multimeter (SDM3055) working simultaneously by means of a custom virtual instrument (VI) developed using LabVIEW (National Instruments, Austin, TX, USA). We increased the force applied in digitally controlled, high-resolution steps of 0.1–0.5 N while recording resistance. After force calibration testing, force was plotted against resistance for each sensor used and a polynomial equation was fitted so that force data were provided from the measured change in resistance.

#### 2.1.3. Data Preprocessing

At times, we encountered connectivity issues with the sensors, causing them to disconnect during the surgical task. In a few other cases, the sensors malfunctioned and could not accurately record the true forces. As a result, we had to remove those problematic trials, which left us with fewer than 20 attempts for some surgeons. We also replaced any negative force values with zeros and applied standard scaling to preprocess the data before using them in our models. This step effectively eliminated any instability introduced by the sensor, ensuring that it does not affect the performance of the models. Examples of force data captured from an expert and a novice surgeon are shown in Figure 3. Our initial experimentation also included a Butterworth filter to cut off high frequencies (where noise is observed), but the performance was similar. Therefore, we excluded it to make the preprocessing step simpler.

### 2.2. Classification Models

#### 2.2.1. Long Short-Term Memory

Long Short-Term Memory (LSTM) is a famous variant of RNN specifically designed for processing and mode sequential data. With its input, output and forget gates, LSTM achieves a remarkable ability to deal with the vanishing gradient problem. In this work, we construct a ‘LSTM’ network, which consists of four LSTM layers with 64 hidden states and one linear layer. After modeling the input time-series force data with a focus on long-term dependencies using the four LSTM layers, the final classification results are obtained through the linear layer.

#### 2.2.2. Bidirectional Long Short-Term Memory

Bidirectional Long Short-Term Memory (Bi-LSTM) comprises two independent LSTM units, one dedicated to forward processing (from the beginning to the end of the sequence) and the other to backward processing. This distinctive architecture facilitates the concurrent management of both historical and prospective information, empowering the network to acquire and leverage contextual insights from past and future temporal directions. The ‘Bi-LSTM’, composed of four layers of bidirectional LSTM with 64 hidden states, is constructed for our experiments.

#### 2.2.3. Gate Recurrent Unit

The Gated Recurrent Unit (GRU) is a gating mechanism that can be targeted as a variant of the LSTM without an output gate. Compared to LSTM, GRU is recognized for its lower parameter and memory demands, leading to decreased computational complexity. This characteristic renders it especially well-suited for deployment in embedded devices and mobile applications. Similarly to ‘LSTM’, we build four-layer deep GRUs with 64 hidden states and a linear layer, denoted as ‘GRU’.

#### 2.2.4. Convolutional Long Short-Term Memory Deep Neural Network

Convolutional Long Short-Term Memory Deep Neural Network (CLDNN) combines the inductive bias of CNNs and the long-term modelling capabilities of LSTM networks to extract features more effectively from a space and time perspective. Consequently, CLDNN has achieved notable success in audio and signal domains. However, CLDNN remains to be explored for force-based neurosurgical skill classification tasks. To this end, we design a CLDNN, denoted as ‘CLDNN’, comprising two Conv1D-ReLU-MaxPool1D layers, followed by four LSTM layers and two linear layers. Detailed architecture is illustrated in Figure 4, and the dimensions of output features $f^{\left(1\right)},f^{\left(2\right)},f^{\left(3\right)},f^{\left(4\right)},f^{\left(5\right)},f^{\left(6\right)},f^{\left(7\right)},f^{\left(8\right)}$ are (64, T/2), (128, T/4), (T/4, 64), (T/4, 64), (T/4, 64), (T/4, 64), (64), (64), respectively.

#### 2.2.5. Temporal Convolution Network

Temporal Convolutional Networks (TCNs) offer an efficient way to capture long-range dependencies in time-series data due to their 1D kernels, while they keep the computational costs manageable. They are often favored by researchers because they tend to be more interpretable than other models. TCNs have also shown promising performance in surgical skill assessment tasks [40,41], as well as in surgical activity recognition [42], which is a task closely related to skill assessment.

In this paper, we employ the TCN originally proposed by Anastasiou et al. [40] and adapt it for our task. As illustrated in Figure 5, we use standard 1D convolutions (four layers with output dimensions of $\left\{64,32,16,16\right\}$; kernel size = 25; and stride = 1) followed by 1D max pooling (kernel size = 3, stride = 1) and batch normalization layers. The feature vector with dimensions $F^{\left(4\right)}\times T$ is then averaged across the temporal dimension and fed to a linear layer with a sigmoid activation function to predict surgical skill.

#### 2.2.6. Transformer Network

Transformer networks are known for their ability to capture complex global dependencies in sequence data due to the self-attention mechanism. Although Transformers were originally designed for natural language processing tasks, they have been adapted to time-series modeling, with recent applications in the surgical domain [40,42,43,44].

In this work, we design a simple Transformer network with the following structure, as illustrated in Figure 6. To start with, the force data, represented as a matrix with dimensions of 1×T, are first passed through a 1D convolutional layer. This layer has an output dimension of 16, a kernel size of 1, and a stride of 1, effectively encoding the data into a higher representation, denoted as $f^{\left(1\right)}$. Next, $f^{\left(1\right)}$ is inputted into a multi-head attention block with 8 heads. This helps in capturing long-range dependencies within the data. The resulting output from the attention block is normalized using batch normalization and subsequently averaged along the temporal dimension. This process results in a new feature vector, $f^{\left(2\right)}$, which has dimensions of $1\times F^{\left(2\right)}$. This vector is fed into a linear layer with an output dimension of 16, and a ReLU activation function. Finally, we apply batch normalization and use another linear layer, this time using a sigmoid activation function, to predict the surgical skill.

### 2.3. Data Augmentation

#### 2.3.1. Fast Fourier Transform

Fast Fourier Transform (FFT) converts time-domain data into a frequency-domain representation, allowing us a more clear discerning of the different frequency components within time-series data. It plays a significant role in the analysis of force data, aiding in the comprehension of mechanical behaviors such as vibration, motion, and deformation [45,46]. Considering the real-valued nature of the force data, we employ the Real-valued Fast Fourier Transform (RFFT) [47] to calculate the one-dimensional discrete Fourier transform of the force sequences. The result of RFFT is a complex valued function of frequency. Subsequently, we combine the obtained real and imaginary components from the RFFT result with the force sequences to form a $3\times \times T^{\prime }$ matrix, where $T^{\prime }$ signifies the temporal dimension, and then this matrix serves as the input training data for the network.

#### 2.3.2. Quantization

Quantization aims to quantize time series to a level set, as shown in Figure 7. Values in a time series are rounded to the nearest level in the level set. Quantization can help improve data quality by removing minor noise in the data. In the field of sensor data, noise is common, and quantization contributes to making the data more stable and reliable. In this study, we randomly quantize the force data to ten level sets with a probability of 50% for data augmentation.

#### 2.3.3. Drift

To introduce additional noise disturbances, we apply gradual and random shifts to the values of time-series force data to perform drift data augmentation, as illustrated in Figure 8. We execute drift operations with a 50% probability and specify a maximum drift range of (0.1, 0.5) with five drift points, signifying the random selection of five points where their values are augmented by additional 10% to 50% based on their original values. These five points function as anchor points, and a spline interpolation function is employed to adjust the values of the remaining data.

#### 2.3.4. Time Warp

Time warp introduces timeline distortions to the input, enhancing model robustness in handling data with diverse time scales and speed variations, as shown in Figure 9. During this augmentation, we randomly select five ranges within the sample to double their speed. This time warp augmentation is executed with a 50% probability.

#### 2.3.5. Gaussian Noise

Adding Gaussian noise to the input aims to introduce diversity to the training data, thus helping with regularization, making the model more robust, and reducing overfitting. We use additive noise generated from a normal distribution with zero mean and a standard deviation of 0.1.

#### 2.3.6. Temporal Jittering

We implemented this augmentation method by breaking down the input training sequences into 30 segments, with each segment comprising 10 consecutive time steps. Subsequently, we changed the order of these segments with random shuffling, as illustrated in Figure 10.

## 3. Experiments and Results

### 3.1. Evaluation Protocol

Given the small size of our dataset, we opted for a cross-validation approach to ensure a fair evaluation of the models. We employed two cross-validation strategies: a random sixfold split and a Leave-One-User-Out (LOUO) split, similar to the approach proposed by Ahmidi et al. [48].

In the random split, we created six folds, ensuring each fold contained an equal number of trials from various surgeons, thus mitigating bias. In contrast, in the LOUO scheme, we structured six folds, with each fold containing all the trials from a single expert surgeon and a single novice surgeon. The data of Expert 2 are merged with those of Expert 7, as these were the ones with the fewest trials. During each iteration, five folds were used for training, and one fold was set aside for testing. The results were then averaged across these folds.

The random cross-validation method assessed the models’ adaptability to different data distributions, thereby minimizing bias. Conversely, LOUO evaluated the models’ generalizability to previously unseen surgeons, a crucial consideration for real-world applications.

For model evaluation, we employed standard classification metrics, which include accuracy, precision, recall, and the F1-score. These metrics are defined as follows:(1)$$accuracy=\frac{TP+TN}{TP+TN+FP+FN},$$
(2)$$precision=\frac{TP}{TP+FP},$$
(3)$$recall=\frac{TP}{TP+FN},$$
(4)$$F1-score=\frac{2TP}{2TP+FP+FN},$$
where TP, TN, FP, and FN are the number of True Positive, True Negative, False Positive, and False Negative values, respectively.

### 3.2. Implementation Details

Experiments are conducted using PyTorch on a GeForce RTX 4090Ti GPU. The maximum training epoch is 100 with a batch size of 32. We utilize categorical cross-entropy as the loss function and adopt the Adam optimizer with a fixed learning rate of 1 × 10${}^{-4}$ for all networks. During the training phase, we utilize sequences consisting of 300 consecutive time steps. When dealing with trials that have more than 300 time steps, we randomly select a 300-step sequence from within the trial’s duration. On the other hand, for trials with fewer than 300 time steps, we pad the sequence with zeros. Therefore, the input force data of one trial is re-sampled to 1 × 300. However, during testing, we retain the original temporal resolution of the trial sequences to ensure that the whole execution is evaluated. As for the labels, we use ’1’ to encode expert surgeons and ‘0’ for novice surgeons. We save the trained model with the minimum validation cross-entropy loss for testing.

### 3.3. Performance Analysis

The results obtained from the various models discussed in Section 2.2 are summarised in Table 1 and Table 2. Notably, our investigation identifies TCN and CLDNN as the top-performing models, achieving impressive accuracy rates. Compared with other networks, TCN achieves the highest accuracy of 97.45% in the random-split scheme, while CLDNN follows with an accuracy of 93.65%. In the LOUO scheme, TCN achieves the second-best accuracy of 88.95%, and CLDNN reaches the highest at 96.19%.

It is worth noting that TCN exhibits superior performance in the random-split cross-validation, surpassing the other models with an accuracy gap of 3.8% points over the second-best model, CLDNN. Conversely, CLDNN demonstrates a significant advantage in the LOUO cross-validation, outperforming TCN by a substantial margin of 7.24% points. Remarkably, these models consistently maintain the best performance across all evaluated metrics and cross-validation schemes.

GRU and LSTM models exhibit the weakest performance in both cross-validation setups. However, an interesting observation is that using a Bi-LSTM architecture significantly enhances performance in the LOUO scheme. This highlights the importance of bidirectional data flow in capturing long-range dependencies within force data.

While the Transformer network outperforms the GRU, LSTM, and Bi-LSTM models, it is still worse than the TCN and the CLDNN in all but one metric (precision). Transformer’s performance appears to be limited by data scarcity, aligning with the established understanding that Transformers benefit from larger datasets.

Notably, TCN stands out with the smallest standard deviation among all models in the random split scheme, indicating its consistency across the six folds. In contrast, in the LOUO scheme, all models exhibit high standard deviations, signifying substantial performance variability depending on the selected training and testing folds. An exception to this is the CLDNN model, which maintains a small standard deviation of ±3.22% for accuracy (second best at ±9.94%), ±4.36% for the F1-score (second best at ±14.63%), ±3.73% for precision (second best at ±11.34%), and ±8.24% for recall (second best at ±17.54%).

In Figure 11 and Figure 12, we provide the confusion matrices for the TCN and CLDNN models, respectively. The top row of both figures displays the confusion matrices for the random split cross-validation scheme, highlighting the best-performing fold on the left and the worst-performing fold on the right. In contrast, the bottom row presents the same information but for the LOUO scheme.

An interesting observation is that in the random split scheme, for both models, the False Positive (FP) and False Negative (FN) rates are almost evenly balanced. However, in the LOUO scheme, particularly for the TCN model, the FN rates are noticeably higher. This discrepancy indicates that the models tend to misclassify expert trials as novice trials more frequently in the LOUO scheme.

It is crucial to emphasize that while TCN demonstrates superior performance in the random split scheme, CLDNN’s performance, as indicated by multiple metrics and its consistency in standard deviation, establishes it as the more reliable model. Furthermore, it performs better in the most challenging cross-validation scheme, LOUO, making it more applicable in real-world applications. CLDNN’s success could be attributed to the combination of two distinct sequence modeling techniques, temporal convolutions and LSTMs, offering a more comprehensive approach to utilizing convolutional inductive biases and capturing long-range dependencies, which are essential for surgical skill assessment.

### 3.4. Data Augmentation Analysis

We further investigate the effects of different data augmentations on the two top-performing models, CLDNN and TCN. Table 3 and Table 4 show the model performance with six data augmentation methods for random split and LOUO settings, respectively.

For the random split setting, as depicted in Table 3, TCN timewarp attains the highest mean accuracy at 97.46% with the lowest standard deviation of ±2.56% and the second-highest mean F1-score at 97.40% with the lowest standard deviation of ±2.35% among six folds. It indicates that despite TCN already achieving high performance under the random split setting, time warp augmentation still offers certain benefits, particularly in reducing standard deviation. In addition, CLDNN with FFT augmentation (CLDNN-FFT) achieves a superior F1 score of 93.63 ± 4.24% and maintains a comparable accuracy of 93.24 ± 4.73%. Furthermore, it is worth noting that all data augmentation algorithms, with the exception of Gaussian noise for CLDNN and drift for TCN, contribute to reducing the standard deviation of accuracy and the F1-score. This observation underscores that under the random split setting, data augmentation effectively fortifies the model’s robustness to some degree.

For the LOUO setting, as shown in Table 4, CLDN timewarp achieves the highest overall performance with an accuracy of 96.48 ± 2.74% and an F1 score of 95.89 ± 3.73%. In comparison to CLDNN without augmentation, CLDNN timewarp demonstrates an improvement of 0.29 ± 0.48% in accuracy and 0.35 ± 0.63% in the F1 score. Additionally, quantization yields a modest enhancement in the mean accuracy and the mean F1 score of CLDNN. Furthermore, it is pleasant to observe that all data augmentation methods are beneficial for the average accuracy of TCN, and all but two methods (quantization and temporal jittering) are helpful for the average F1 score of TCN. Compared with TCN without augmentation, TCN-FFT obtains the most significant improvements, demonstrating an accuracy enhancement of 6.01 ± 9.45% and an F1 score improvement of 5.66 ± 7.66%.

It can be seen from Table 3 and Table 4 that under the more challenging and realistic LOUO setting, data augmentation methods show a more significant improvement in model performance compared with the random split setting, which demonstrates the importance of data augmentation in clinical applications.

### 3.5. Qualitative Analysis

To gain deeper insights into the reasoning behind the predictions made by our top-performing models, i.e., CLDNN and TCN, we employ a technique that involves extracting activations from the final layer just before the linear layers for each model. These activations are subsequently superimposed onto the original input sequence, effectively pinpointing areas within the execution where the network exhibits higher attention to inform its predictions. In essence, this process generates feature vectors with dimensions $f\times T^{\prime }$, where *f* represents the feature dimension and $T^{\prime }$ signifies the temporal dimension. In particular, for the TCN model, these activations correspond to the feature vector denoted as $F^{\left(4\right)}$, while for the CLDNN model, they correspond to $F^{\left(7\right)}$. Initially, we average the feature dimension, resulting in a compact vector of dimensions $1\times T^{\prime }$. We then normalize these values to a range between zero and one. Subsequently, this vector is upsampled to align with the original sequence’s temporal resolution *T*.

Following the upsampling stage, we utilize the resulting array to represent color intensity, overlaying it onto the force data plot, as demonstrated in Figure 13. In this visualization, warmer colors signify regions where the network focuses more to make its final prediction, while cooler tones indicate areas of relatively lower network attention.

Figure 13 showcases these visualizations produced for surgical executions conducted by both expert and novice surgeons, using both the TCN and CLDNN models. Specifically, Figure 13a,c illustrate visualizations for an expert surgeon using the TCN and CLDNN models, respectively, while Figure 13b,d display visualizations for a novice surgeon. Notably, in the visualizations for novices, it is apparent that the networks exhibit higher activation in regions characterized by sharp force peaks and sudden turning points. These patterns could be linked to surgical skill, as rapid changes and precise responses may correlate with competency. In contrast, the visualizations for experts reveal that the networks are more activated in regions where lower forces are observed. This observation suggests that the networks are strategically focusing on specific regions within the surgical executions to make skill level predictions.

It is important to note that subtle differences exist between the plots generated by TCN and CLDNN. These can be attributed to variations in the way in which each network computes these activations (1D convolutions vs LSTM). Another factor contributing to the difference is that TCN maintains the original temporal resolution throughout the network, whereas in CLDNN, the feature vector $F^{\left(7\right)}$ undergoes significant downsampling due to pooling layers, resulting in reduced resolution. Consequently, this affects the visualization presented in the final plot.

### 3.6. Limitations

#### 3.6.1. Hardware and Study Limitations

A limitation is related to the piezoresistive sensor. As this is a newly developed device, the sensor might not be optimally placed to measure all forces for all instruments. For instance, one novice surgeon was applying a very light touch on the force sensor throughout the tests, which could clearly be seen when examining the way they were holding the instrument. They were applying the force by exerting pressure on the instrument using the index instead of the thumb, which emphasizes the fact that the design of the sensorized gloves must be customized and adapted to the vast spectrum of techniques and ways each surgeon has of holding the instruments.

Further, our sensor was located at a single point on the surgical glove (Figure 1). In order to solve this issue, we developed new designs of the glove to accommodate for the different ways each clinician holds and positions the microscissors in the right-hand thumb. Over time and through iterative improvement, the sensor placement is being continuously optimized. For example, a body-mounted sensor could be constructed to reduce the need to be connected to a fixed device by wire. There is a concern that some surgeons may be left-handed. As the current gloves are easily customized, it is straightforward to place the sensor on the left-hand glove, enabling data collection from left-handed surgeons. In addition, the next glove iteration could integrate wireless sensors, reduce form factor, and simplify the methods and materials used for sensor-to-glove attachment to make it easier and quicker to reconfigure the device accommodating right-handed and left-handed participants.

Another issue regarding the sensor and its connection to the data acquisition device is the fact that the sensor occasionally disconnects when the surgeon wearing the glove makes sudden ample movements. When this takes place, the cables disconnect and the data for that particular test are not recorded properly. To overcome this, longer cables have been employed for the glove to facilitate surgeons’ operations. Nevertheless, we are in the process of designing a wireless node that can be mounted on the wrist of the glove to send the data wirelessly to the laptop. This solution has the potential to ultimately solve the disconnection problem.

Moreover, for highly invasive and more complex surgeries, the current piezoresistive sensor may omit valuable micro-information due to the surgeon’s force feedback having an even larger and more precise range and fluctuation. To address this, we are working to optimize the piezoresistive conductive material to achieve better reproductivity accuracy and extend the force range. We are also refining the signal acquisition and processing circuitry to reduce noise and latency, thereby obtaining more precise force data, especially in scenarios where the force falls below 3 N.

Additionally, it is unclear whether the profile of the sensors can impair the haptic feedback for the operating surgeon, which could interfere with a real-world operation.

Furthermore, although the grape dissection task was a useful starting point, it is low fidelity and does not simulate complications, which in real life might lead to increased force exertion as the operating surgeon is under pressure. In addition, although our dataset includes 13 surgeons to provide adequate variability for validating our pilot study, it is collected from a single centre. Therefore, further work should use smart gloves in high-fidelity simulation models and collect data from surgeons in different geographical locations/hospitals to increase data diversity and further validate the findings of this study.

Lastly, it is important to acknowledge the limited sample size of our dataset. While this pilot study successfully demonstrates the feasibility of modeling differences between experts and novices using deep learning on a fundamental yet frequently performed microsurgical task, it is important to gather additional data for a more robust validation of our approaches.

#### 3.6.2. Algorithm Limitations

The findings illustrated in Figure 11 and Figure 12 point to the challenges encountered by both models when evaluating the LOUO cross-validation scheme. It becomes apparent that this difficulty is characterized by higher FN rates, indicating a tendency to misclassify surgeons as novices. The complexity of LOUO as a cross-validation scheme lies in its requirement for models to effectively generalize to previously unseen surgeons. Furthermore, in the context of small datasets, as is the case here, the validation samples may diverge from the data distribution observed in the training set. This divergence heightens the likelihood of misclassification errors.

It is worth noting that, excluding CLDNN, all other models exhibit an additional limitation in their LOUO results, which are detailed in Table 2. These models exhibit a large standard deviation across different folds, emphasizing their sensitivity to the specific distribution of training and validation data within each fold. This dependency on data distribution introduces variability in their performance, further complicating the LOUO evaluation process.

Another potential limitation is that not all data augmentation methods bring significant performance improvements to deep learning models for FSC in microsurgery. While the main reason is that the proposed models already achieve high performance, thereby leaving little room for data augmentation to take effect, it remains crucial to acknowledge the essential role of choosing appropriate data augmentation techniques when utilizing force data for automated surgical skill classification. According to our data augmentation analysis in Section 3.4, time warp and FFT augmentations could be given priority and considered for implementation.

Lastly, although, to the best of our knowledge, there are no gold standard tools to validate the classification result of our models, we can refer to the work by Horsfall et al. [35], where they showed that for the same dataset, there was a significant difference (*p* = 0.002) between the median forces applied by experts and novices.

## 4. Conclusions

This work aims to address FSC tasks in microsurgery. To investigate great deep learning models and data augmentation techniques for force data, we construct a novel FSC dataset, which is collected by sensorized surgical gloves. Based on this dataset, we explore and summarize six state-of-the-art deep learning model frameworks (LSTM, Bi-LSTM, GRU, CLDNN, TCN, and Transformer) and six data augmentation algorithms (FFT, quantization, drift, time warp, Gaussian noise, and temporal jittering) through comprehensive quantitative, qualitative, and visual analyses. Two different types of cross-validation schemes, i.e., random split and LOUO, are implemented to provide a full evaluation of model performance. For the random split setting, TCN outperforms the other five deep learning models in all four metrics (accuracy, F1-score, precision, and recall). Furthermore, the implementation of time warp data augmentation with TCN results in an additional accuracy improvement, reaching 97.46%, and reduces standard deviations across all four metrics. For the LOUO setting, CLDNN excels in all four metrics with values of 96.19%, 95.54%, 98.33%, and 93.44%. Time warp augmentation further improves the accuracy and the F1-score of CLDNN by 0.29% and 0.35%, respectively, while reducing the standard deviations for all metrics.

Our experimental results suggest that CLDNN and TCN are well-suited for FSC tasks, with time warp being a preferable choice when considering data augmentation techniques. The FSC algorithms developed in this study have the potential to bring significant benefits to microsurgery training and practice, particularly in terms of real time, objectivity, and transferability.

Although microsurgery can be complicated, we believe that force data alongside other data inputs (e.g., video) embed valuable information on surgeon performance that can be used to develop learning-based models and derive discriminative features for automated skill assessment. Our study is a pilot to investigate this hypothesis, revealing that differences in the force exerted between experts and novices during a simple yet routinely performed microsurgical suturing task can indeed be modeled with deep learning architecture. Ultimately, this is validated by the obtained accuracy rates and F1-scores surpassing 95%. Future research will focus on utilizing multimodal data for automated surgical skill assessment.

## Figures and Tables

**Figure 1 sensors-23-08947-f001:**
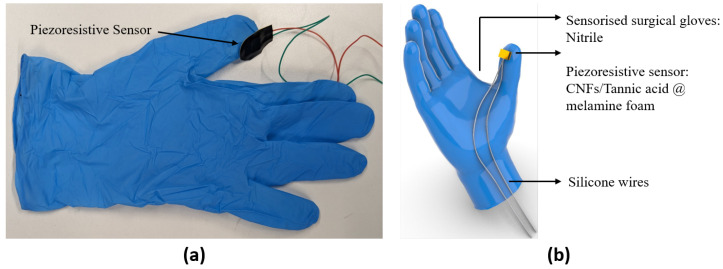
(**a**) Prototype of the sensorized surgical glove. (**b**) CAD model of the sensorized surgical glove.

**Figure 2 sensors-23-08947-f002:**
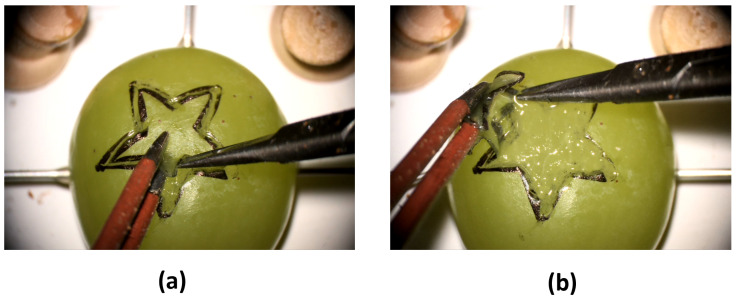
(**a**,**b**) Demonstration of the validated dissection task ’Star’s the limit’ with dissection of the grape skin in the shape of a star.

**Figure 3 sensors-23-08947-f003:**
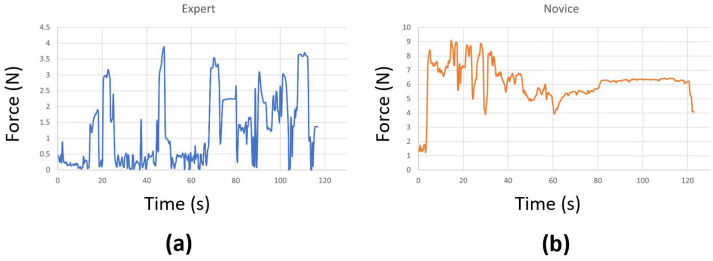
Force data from (**a**) an expert and (**b**) a novice surgeon.

**Figure 4 sensors-23-08947-f004:**
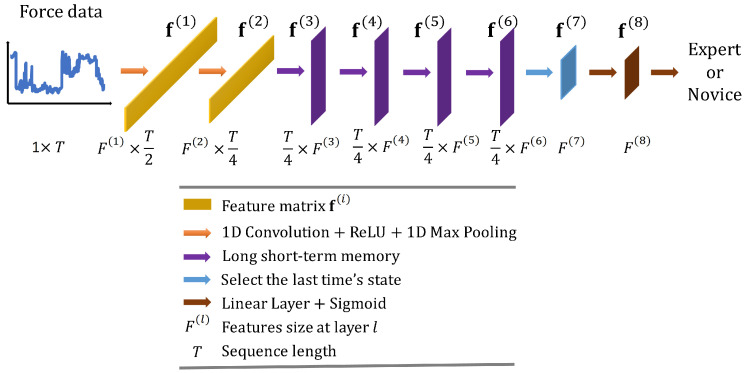
Overall architecture of the proposed convolutional long short-term memory deep neural network ‘CLDNN’.

**Figure 5 sensors-23-08947-f005:**
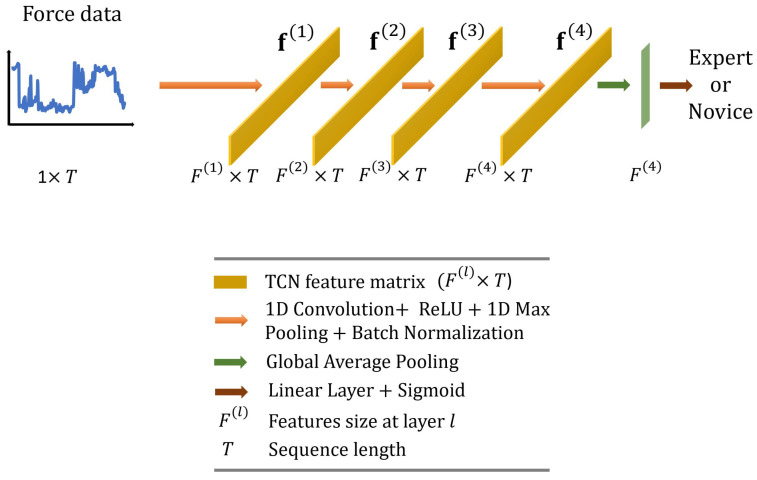
Overall architecture of the proposed temporal convolutional network.

**Figure 6 sensors-23-08947-f006:**
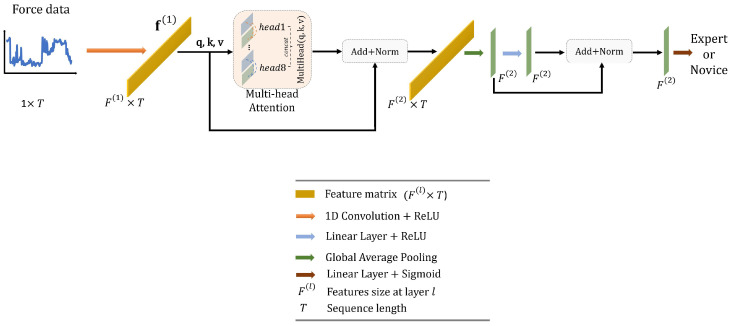
Overall architecture of the proposed Transformer.

**Figure 7 sensors-23-08947-f007:**
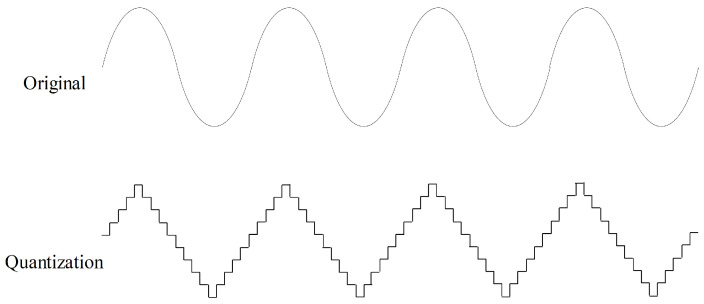
Quantization data augmentation with ten level sets.

**Figure 8 sensors-23-08947-f008:**
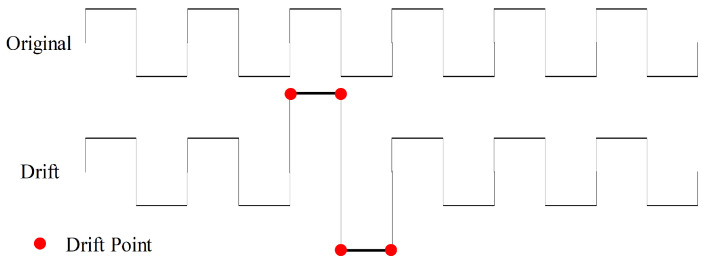
Drift data augmentation with four drift points.

**Figure 9 sensors-23-08947-f009:**
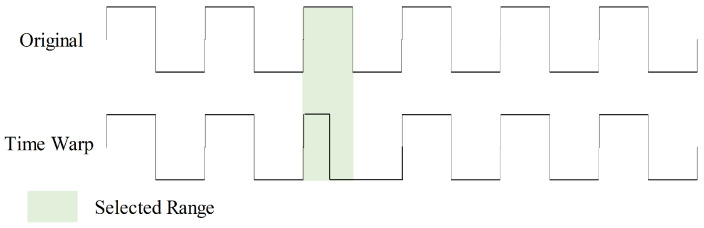
Time warp data augmentation with one selected range.

**Figure 10 sensors-23-08947-f010:**
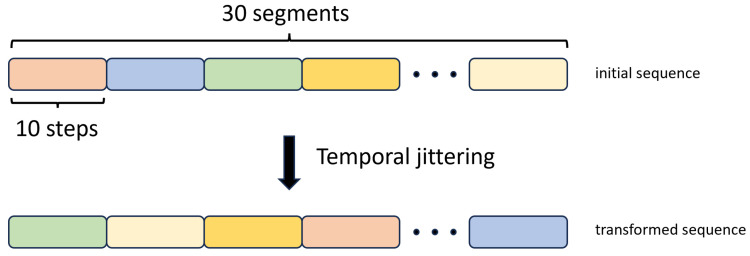
Temporal Jittering data augmentation method.

**Figure 11 sensors-23-08947-f011:**
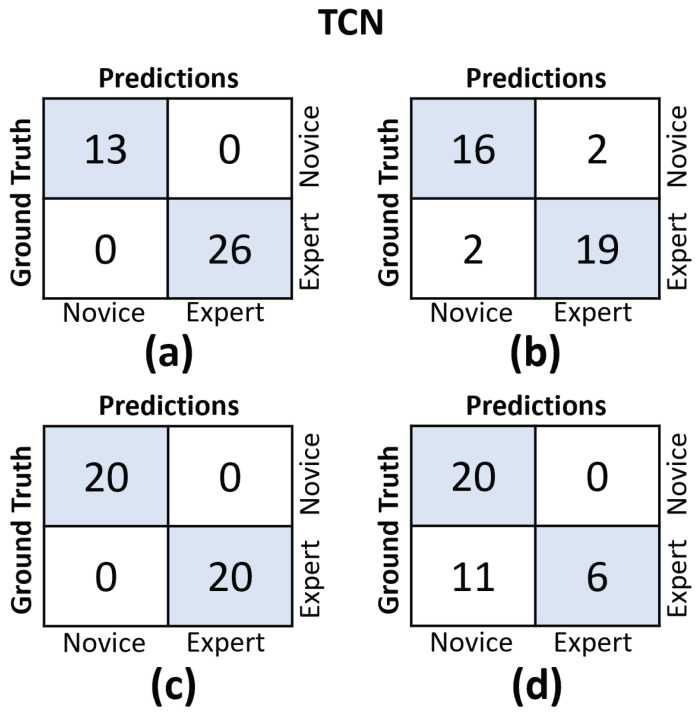
Confusion matrices for the TCN model. (**a**) Results on the best-performing fold of the random split. (**b**) Results on the worst-performing fold of the random split. (**c**) Results on the best-performing fold of the LOUO cross-validation scheme. (**d**) Results on the worst-performing fold of the LOUO cross-validation scheme.

**Figure 12 sensors-23-08947-f012:**
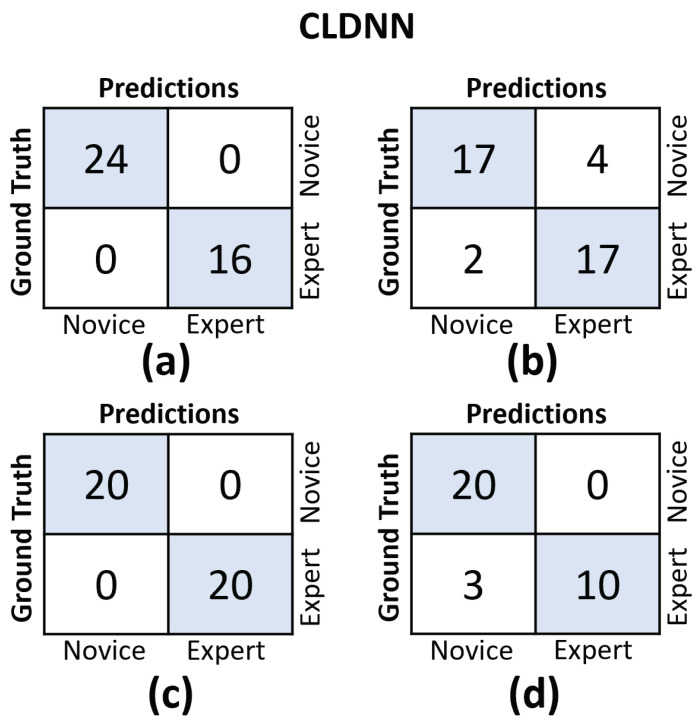
Confusion matrices for the CLDNN model. (**a**) Results on the best-performing fold of the random split. (**b**) Results on the worst-performing fold of the random split. (**c**) Results on the best-performing fold of the LOUO cross-validation scheme. (**d**) Results on the worst-performing fold of the LOUO cross-validation scheme.

**Figure 13 sensors-23-08947-f013:**
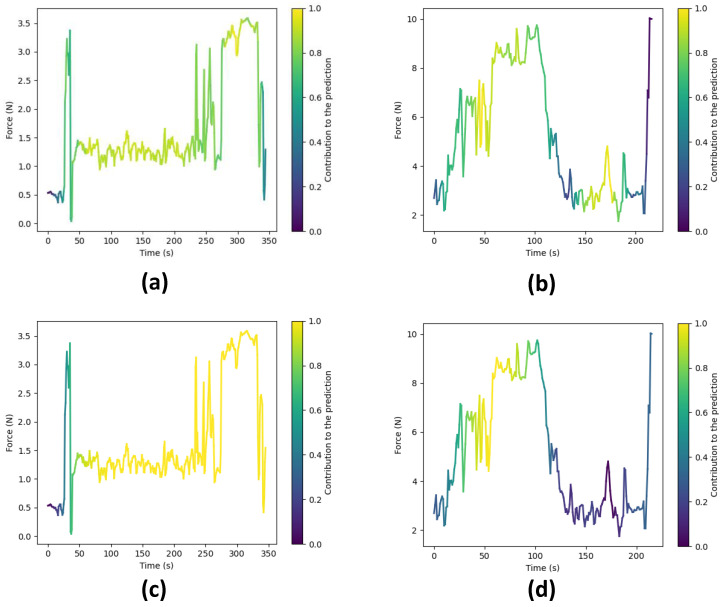
Force plots showing the areas where the network focuses to make a prediction. Warmer colors signify higher attention. (**a**) Expert surgeon predicted as ‘Expert’ by the TCN. (**b**) Novice surgeon predicted as ‘Novice’ by the TCN. (**c**) Expert surgeon predicted as ‘Expert’ by the CLDNN. (**d**) Novice surgeon predicted as ‘Novice’ by the CLDNN.

**Table 1 sensors-23-08947-t001:** Classification results for all models on the random split cross-validation scheme. Results are averaged across all 6 folds. The best two results are in **bold** and underlined.

Method	Accuracy	F1-Score	Precision	Recall
GRU	75.46 ± 5.05	76.08 ± 4.17	73.96 ± 9.74	80.36 ± 8.74
LSTM	78.86 ± 7.40	76.91 ± 5.30	88.58 ± 16.58	71.15 ± 9.90
Bi-LSTM	80.51 ± 4.06	77.98 ± 4.42	87.45 ± 7.49	71.72 ± 9.43
CLDNN	93.65 ± 5.39	93.51 ± 5.29	92.63 ± 6.84	94.56 ± 4.44
TCN	**97.45** ± 3.63	**97.51** ± **3.37**	**97.54** ± **3.70**	**97.54** ± **3.70**
Transformer	90.67 ± **3.20**	89.77 ± 3.93	94.71 ± 6.21	85.71 ± 5.29

**Table 2 sensors-23-08947-t002:** Classification results for all models on the LOUO cross-validation scheme. Results are averaged across all 6 folds. The best two results are in **bold** and underlined.

Method	Accuracy	F1-Score	Precision	Recall
GRU	77.57 ± 10.88	74.55 ± 15.91	78.62 ± 13.30	76.97 ± 22.72
LSTM	76.67 ± 14.33	72.43 ± 21.46	78.50 ± 14.50	74.56 ± 26.36
Bi-LSTM	84.92 ± 9.94	80.65 ± 15.06	90.39 ± 11.34	76.37 ± 21.24
CLDNN	**96.19** ± **3.22**	**95.54** ± **4.36**	**98.33** ± **3.73**	**93.44** ± **8.24**
TCN	88.95 ± 14.80	88.44 ± 14.63	90.00 ± 18.26	92.16 ± 17.54
Transformer	86.68 ± 13.37	83.42 ± 19.73	88.2 ± 16.11	86.95 ± 25.89

**Table 3 sensors-23-08947-t003:** Classification results for all data augmentations on the random split cross-validation scheme. Results are averaged across all 6 folds. The best two results are in **bold** and underlined.

Method	Accuracy (%)	F1-Score (%)	Precision (%)	Recall (%)
CLDNN (no augmentations)	**93.65** ± 5.39	93.51 ± 5.29	92.63 ± 6.84	94.56 ± 4.44
CLDNN-FFT	93.24 ± 4.73	**93.63** ± 4.24	91.21 ± 6.69	**96.49** ± **3.74**
CLDNN-drift	92.36 ± **2.12**	92.03 ± **2.68**	91.70 ± 6.54	92.88 ± 4.43
CLDNN-quantize	93.23 ± 4.48	93.43 ± 4.16	92.46 ± 8.08	95.03 ± 4.00
CLDNN-timewarp	93.21 ± 3.15	92.81 ± 3.65	**96.52** ± **3.64**	89.67 ± 6.33
CLDNN-Gaussian noise	92.79 ± 4.94	92.50 ± 5.39	93.13 ± 7.45	92.99 ± 9.37
CLDNN-temporal jittering	91.12 ± 3.11	90.73 ± 3.42	90.28 ± 7.25	91.87 ± 5.70
TCN (no augmentations)	97.45 ± 3.63	**97.51** ± 3.37	97.54 ± 3.70	**97.54** ± 3.70
TCN-FFT	95.75 ± 2.85	95.39 ± 2.85	96.73 ± 4.64	94.28 ± 3.65
TCN-drift	97.02 ± 3.75	97.04 ± 3.56	97.49 ± 3.73	96.61 ± 3.66
TCN-quantize	96.61 ± 2.83	96.35 ± 2.93	97.56 ± 3.60	95.32 ± 4.20
TCN-timewarp	**97.46** ± **2.56**	97.40 ± **2.35**	**97.61** ± **3.56**	97.29 ± **2.75**
TCN-Gaussian noise	97.45 ± **2.56**	97.40 ± **2.35**	**97.61** ± **3.56**	97.29 ± **2.75**
TCN-temporal jittering	95.33 ± 3.12	94.88 ± 3.04	96.45 ± 3.76	93.48 ± 3.87

**Table 4 sensors-23-08947-t004:** Classification results for all data augmentations on the LOUO cross-validation scheme. Results are averaged across all 6 folds. The best two results are in **bold** and underlined.

Method	Accuracy (%)	F1-Score (%)	Precision (%)	Recall (%)
CLDNN (no augmentations)	96.19 ± 3.22	95.54 ± 4.36	**98.33** ± 3.73	93.44 ± 8.24
CLDNN-FFT	90.19 ± 7.77	88.29 ± 10.09	93.47 ± 10.05	86.55 ± 16.39
CLDNN-drift	94.59 ± 5.42	94.21 ± 5.38	94.44 ± 9.21	94.70 ± 5.32
CLDNN-quantization	96.36 ± 4.43	95.56 ± 5.72	97.44 ± 5.73	93.92 ± 6.88
CLDNN-timewarp	**96.48** ± **2.74**	**95.89** ± **3.73**	97.73 ± **3.33**	94.22 ± 5.21
CLDNN-Gaussian noise	88.67 ± 9.47	86.68 ± 12.47	90.78 ± 13.87	87.66 ± 18.97
CLDNN-temporal swapping	92.39 ± 6.45	92.40 ± 6.77	88.35 ± 10.44	**97.44** ± **3.63**
TCN (no augmentations)	88.95 ± 14.80	88.44 ± 14.63	90.00 ± 18.26	92.16 ± 17.54
TCN-FFT	**94.96** ± **5.35**	**94.10** ± **6.97**	**94.84** ± **7.33**	93.72 ± **8.59**
TCN-drift	89.46 ± 13.85	88.73 ± 14.20	90.33 ± 17.53	92.16 ± 17.54
TCN-quantization	89.57 ± 13.06	87.55 ± 16.36	91.09 ± 15.88	90.20 ± 21.92
TCN-timewarp	90.36 ± 13.26	90.29 ± 12.37	90.33 ± 17.53	**94.12** ± 13.15
TCN-Gaussian noise	90.59 ± 10.88	88.55 ± 14.11	91.48 ± 13.70	89.89 ± 19.36
TCN-temporal jittering	89.32 ± 12.74	86.62 ± 17.82	91.52 ± 14.95	88.74 ± 23.93

## Data Availability

The code and dataset for this study are available at https://doi.org/10.5522/04/24476641 (accessed on 29 September 2023).

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
