# Peer review of "A Deep Learning Approach to Classify Surgical Skill in Microsurgery Using Force Data from a Novel Sensorised Surgical Glove"

_sensors, 2023, doi:10.3390/s23218947_

Round 1

Reviewer 1 Report

Comments and Suggestions for Authors

Final decision: Manuscript is acceptable for publication after minor correction.

Abstract: Current abstract is acceptable

Keywords: Acceptable

Introduction: Acceptable but could be better by wider literature review.

Introduction:
The introduction section should be briefly highlight the more importance and necessity of this study

M & M: Current M & M is acceptable.

Bibliography/References:  Authors should do more literature review for using related articles.

 I could not see running title.

Final decision: Manuscript is acceptable for publication after minor correction as recommended 

Reviewer 2 Report

Comments and Suggestions for Authors

Strengths

1.     Abstract: Explained well with the aims and details of the project. Includes a brief description of the methods used and the results obtained. 

2.     Lines 42-32: A clear and good connection between the background research and the current topic in question. Provides good flow and relates back to the aim of the research continuously. 

3.     Lines 68: The author mentions that this new surgical glove will be low cost, which relates back to the current cost issue in microsurgery. 

4.     Lines 97-99: Shows a good understating of current related studies and identifies limitations with these techniques. 

5.     1.3 Contributions: Identifies clearly the main purpose of the paper, including the number of trials and surgeons used, as well as clearly outlining the data models used in the analysis of this research. 

6.     Lines 218-219 and lines 224-226: A good comparison of methods and models according to the needs and requirements of the research project. 

7.     Lines 282-304: Good use of figures with relevant text 

8.     Lines 350-352: Excellent validation of analysis methods, shows the researcher has a good insight in what methods are required to achieve the correct and relevant results according to the research topic. 

9.     Lines 460-461 : The authors explained a new novel idea for progression of the current device. 

10.  Lines 465-466: Good solution found to an issue the researcher previously mentioned in the method of the experiment. 

11.  Lines 515-518: The author clearly states that the new novel device has a significant positive impact on the betterment of microsurgery training and practice. This is proven in the above results. 

Corrections and Comments 

1.     Line 37: Add “and” between the words “model towards” to allow for fluidity when reading. 

2.     Line 61: Paragraph should end with a reference as a paragraph starts. 

3.     Lines 72-73: The author mentions the use of Machine Learning as a KEY differentiating factor between their research topic and other research done before, as other researchers have also used force data in microsurgery (without ML or DL). It may help to mention Machine Learning or Deep Learning in the title of the topic; however, it is up to the author as they have mentioned it in the abstract of the paper. 

4.     Line 88: The author mentions that force data is an effective measure of skill in minimally invasive surgery. However, they may like to speak about the impact of force data in highly invasive or more complex surgery in their limitation/discussion at the end of the paper, when discussing future progression of the device. 

5.     Line 90, 93, 103, 105, 109, 112: The author switches from using number referencing to the name of the reference within the text as a referencing style. The author must keep to a consistent referencing system. 

6.     Line 91, 37, 309: The author uses number referencing as part of the sentence which results in a break of the reading flow and confuses the reader. The author should use the title or the name of the reference and then add the number reference after or at the end of the sentence. 

7.     Lines 118-119: The author capitalises and adds bold to the first letter of each word used in the abbreviation. Bold is not necessary as the author uses only capitalisation consistently in the rest of the paper. If the author would also like to use bold, it must be consistent for all abbreviation titles in the paper. 

8.     Line 137: Surgeons from a single tertiary hospital were used in this research project. The author may like to comment on the validity of this in the limitations and whether using surgeons from different geographical points may have impacted the results or increased the validity of the data. 

9.     Line 198: Figure 3 is mentioned; however, the figure is under 2.2.1 Long Short-Term Memory where there is no further mention of Figure 3. It should be moved next to the relevant text.

10.  Table 1 & 2: The table has been inserted in the middle of a paragraph where there is no mention of the table. The tables are mention in the next section 3.3; hence they should be placed with the relevant text. 

11.  Line 407: Tables 3 & 4 are mentioned however, they are shown in the next section 3.5 where this is not mention of said tables. Tables should be places with the relevant text. 

12.  Line 456: The author mentions Figure 2d. There is no such figure in the research paper. 

13.  Line 458: The author mentions the device was places on a right-handed glove thumb. The author may want to address another limitation that some surgeons may be left-handed. What can be done in the future progression of the device to solve this issue? 

14.  Line 509: Missing full stop between “accuracy Furthermore,”. 

Comments on the Quality of English Language

Clear and concise with good use of grammar and scientific terminology. Remember punctuation and connecting words, 

Reviewer 3 Report

Comments and Suggestions for Authors

Xu et al. reported a combination of a wearable piezoresistive force sensor and a deep learning model to classify the surgical skill levels. This research is interesting and could potentially reveal some opportunities for AI and/or flexible sensors in promoting microsurgery practice.  Still, several questions:

1. As the author uses a homemade force sensor to validate the the machine learning models, any variation/instability/sensitivity of the customized sensor may significantly affect the results/accuracy. 

2. The sample size is still small, and the results may not support a larger amount of clinicians. 

3. The experimental setup is too simple and may induce more noise from motion artifacts or extended wires. 

4. Are there any gold-standard tools or more validation groups to further confirm the modeling accuracy? The final target for this work is to differentiate novices from experts and help the novice find their inappropriate actions during the surgery. More data needs to be provided to prove the concept. 

5. Surgeries, especially microsurgery, should be very complicated, and the reviewer questions if a force sensor is enough to classify the surgical skills. 

Round 2

Reviewer 3 Report

Comments and Suggestions for Authors

The authors have already answered the comments one by one with detailed explanations and sufficient experimental results. The revised version is well-written, well-organized, and improved a lot compared with the previous version. Therefore, the reviewer recommends to accept.